# Evaluation of Automatic Segmentation of Thalamic Nuclei through Clinical Effects Using Directional Deep Brain Stimulation Leads: A Technical Note

**DOI:** 10.3390/brainsci10090642

**Published:** 2020-09-17

**Authors:** Marie T. Krüger, Rebecca Kurtev-Rittstieg, Georg Kägi, Yashar Naseri, Stefan Hägele-Link, Florian Brugger

**Affiliations:** 1Department of Neurosurgery, Cantonal Hospital, 9000 St. Gallen, Switzerland; yashar.naseri@kssg.ch; 2Department of Stereotactic and Functional Neurosurgery, University Medical Center, 79106 Freiburg, Germany; 3R&D Functional and Stereotactic Neurosurgery, Brainlab AG, 81829 Munich, Germany; rebecca.rittstieg@brainlab.com; 4Department of Neurology, Cantonal Hospital, 9000 St. Gallen, Switzerland; georg.kaegi@kssg.ch (G.K.); stefan.haegele-link@kssg.ch (S.H.-L.); florian.brugger@kssg.ch (F.B.)

**Keywords:** automatic anatomical segmentation, deep brain stimulation, directional leads, essential tremor, GUIDE XT, ventral intermediate nucleus

## Abstract

Automatic anatomical segmentation of patients’ anatomical structures and modeling of the volume of tissue activated (VTA) can potentially facilitate trajectory planning and post-operative programming in deep brain stimulation (DBS). We demonstrate an approach to evaluate the accuracy of such software for the ventral intermediate nucleus (VIM) using directional leads. In an essential tremor patient with asymmetrical brain anatomy, lead placement was adjusted according to the suggested segmentation made by the software (Brainlab). Postoperatively, we used directionality to assess lead placement using side effect testing (internal capsule and sensory thalamus). Clinical effects were then compared to the patient-specific visualization and VTA simulation in the GUIDE™ XT software (Boston Scientific). The patient’s asymmetrical anatomy was correctly recognized by the software and matched the clinical results. VTA models matched best for dysarthria (6 out of 6 cases) and sensory hand side effects (5/6), but least for facial side effects (1/6). Best concordance was observed for the modeled current anterior and back spread of the VTA, worst for the current side spread. Automatic anatomical segmentation and VTA models can be valuable tools for DBS planning and programming. Directional DBS leads allow detailed postoperative assessment of the concordance of such image-based simulation and visualization with clinical effects.

## 1. Introduction

In deep brain stimulation (DBS), precise surgical planning of the target is one of the most crucial steps impacting patient outcomes. In particular for standard DBS targets used in movement disorders, such as the subthalamic nucleus (STN), ventral intermediate nucleus of the thalamus (VIM), and the internal globus pallidus (GPi), it has become evident that millimeters matter. While Nickl et al. reported significantly improved motor outcomes in STN-DBS in Parkinson’s disease patients after surgical lead revision, with a median displacement distance of 4.1 mm (range 1.6–8.42 mm) [1], Akram et al. had a 1.5 mm targeting error in DBS, beyond which they routinely relocated implanted leads [2].

For targeting, there are two basic approaches: indirect targeting using coordinates derived from autopsy-based atlases, or direct visualization of target structures on magnetic resonance imaging (MRI). However, in contrast to the STN and GPi, in vivo segmentation of thalamic nuclei such as the VIM remains challenging, despite significant advances in neuroimaging technology [3,4]. Although some groups have reported VIM visualization on 3T proton density [5], fast gray matter T1 inversion recovery sequences [6], or ultra-high field MRI [4], it is not readily visible on conventional stereotactic MR imaging sequences. Other recent advances include connectivity-based segmentation of the VIM using probabilistic tractography [2]. This approach is, however, not yet commercially available.

Thus, in most institutions, VIM targeting remains an indirect coordinate-based approach in relation to the anterior commissure (AC) and posterior commissure (PC), along with other identifiable structures such as the lateral thalamic/internal capsule border [5,6] and the wall of the third ventricle. While adjustments can be made through intraoperative clinical testing in awake surgeries, precise targeting and the use of high-quality imaging is even more important for asleep DBS without this option.

Recently, advanced automatic segmentation methods have become available in CE-marked clinical planning software. Although they do not replace traditional DBS targeting, they can support both preoperative planning and postoperative programming through 3-dimensional (3D) visualization of leads in their spatial relationship to target structures. This 3D perception is gaining importance with the advent of directional DBS leads, in particular when integrating volume of tissue activated (VTA) models for an easier understanding of the effects and side effects of stimulation. However, any anatomical segmentation needs to prove accurate and patient-specific to be clinically useful. Reinacher et al. therefore compared the automatic STN segmentation available in Brainlab Elements^®^ (Brainlab AG, Munich, Germany) to the gold standard of electrophysiology for 175 trajectories; they found it to be a viable option to support planning, with an overall median deviation of 1.1 mm for STN entry and 2.0 mm for STN exit [7].

Here, we investigate the performance of automatic segmentation of thalamic nuclei using Brainlab Elements, as well as VTA modeling with Guide™ XT (Boston Scientific, Natick, MA, USA) in a case of asymmetric patient anatomy. We compare the image-based visualization and simulation with the patient’s clinical stimulation effects through current steering towards the internal capsule and sensory thalamus using directional DBS leads.

## 2. Materials and Methods

### 2.1. Patient

A 58-year-old man with bilateral essential tremor received bilateral VIM-DBS with directional leads (Model DB-2202-45; Boston Scientific). Due to his asymmetrical anatomy, also highlighted by the surgical planning software, the right trajectory was adjusted by 2 mm in the anterior direction (Figure 1). The patient experienced excellent tremor control on both sides, with no side effects on his latest stimulation settings. Six months postoperatively, he participated in this study to test tremor control and side effects in a double-blinded fashion.

### 2.2. Preoperative Planning

Pre-planning was performed the day before surgery using Elements Stereotaxy (Brainlab AG, Munich, Germany). MR imaging data was acquired under general anesthesia 14 days before surgery on a 3T scanner (Magnetom Skyra, Siemens) and included the following sequences: T1-weighted MPRAGE with contrast agent (0.9 mm slice thickness), axial T2 STIR (2 mm slice thickness), sagittal T2 SPACE (1 mm slice thickness), and axial SWI (1.8 mm slice thickness). All sequences were loaded into the software and rigidly co-registered, after which automatic segmentation of the basal ganglia was performed using Elements Anatomical Mapping. The respective algorithm is based on a patented Synthetic Tissue Model, which adapts in a flexible manner to the respective image modality and detected sub-modality. This unique approach therefore enables the simultaneous non-linear registration of up to six different MR sequences to yield a consistent, patient-specific segmentation. The VIM, ventral posterior lateral nucleus (VPL), and ventral posterior medial nucleus (VPM) in Brainlab’s Synthetic Tissue Model are defined according to the Schaltenbrand–Wahren Atlas, which uses Hassler’s nomenclature [8].

Trajectories were then planned on the raw MRI data in Elements Stereotaxy by defining the anterior commissure/posterior commissure (ACPC) and using the following midcommissural point coordinates (ACPC distance = 24.48 mm): x = 14 mm lateral, y = 6.12 mm posterior, z = 0. The subsequent overlay of the automatic segmentation results highlighted an asymmetry in the patient’s anatomy between the left and right hemisphere, with the right VIM displayed more anteriorly. After careful review of visible landmarks and structures on all available sequences, we decided to adjust the planned target point on the right side by 2 mm towards the patient’s anterior, as illustrated in Figure 1D.

### 2.3. Surgical Procedure

On the day of surgery, a stereotactic CT scan was performed (0.7 mm slice thickness, Somatom Force, Siemens, Munich, Germany) after placing a Leksell frame (Leksell^®^ G-frame, Elekta, Stockholm, Sweden), and fused to the preoperatively-acquired MRI scans. Surgery was carried out under local anesthesia. Intraoperative test stimulation up to 5 mA using a macroelectrode (Inomed, Emmendingen, Germany) revealed similar clinical results in both hemispheres for tremor control, as well as similar results for sensory and motor side effects, despite the more anteriorly positioned right lead. The final DBS leads (Vercise Cartesia^TM^ Model DB-2202-45; Boston Scientific, Natick, MA, USA) were therefore implanted as planned without any intraoperative adjustments in terms of position, and with the distal row of segmented electrodes at the target point. The patient received a postoperative CT scan immediately after surgery for postoperative lead assessment (0.7 mm slice thickness, Somatom Force, Siemens, Munich, Germany) and to rule out any complications. The postoperative CT was loaded into Brainlab Elements and co-registered to all preoperative scans. The tip and axis of the implanted DBS leads were automatically detected in Elements Lead Localization and visually verified. After assigning the Vercise Cartesia^TM^ directional lead model in the software, each lead’s orientation was determined automatically from the CT using the intended implantation orientation (in this case anterior) as input and reconfirmed by a rotational fluoroscopy [9].

### 2.4. Postoperative Clinical Testing

Six months after the surgery, the patient returned to our clinic for a blinded DBS programming session. His tremor was well controlled at that time with the following settings: left lead: Case+, 5/6/7-(33%), 60 μs, 130 Hz, 1.5 mA; right lead: IPG+, 13-(100%), 60 μs, 130 Hz, 1.0 mA. He scored 0/32 for the unilateral tasks (rest, posture, intention, handwriting, drawing, and water pouring) of the Fahn Tolosa Marin Tremor rating scale for his right and 0/32 for his left side (before surgery: right side: 14/32; left side 10/32) [10].

The clinician (FB) was blinded to the position and rotation of the leads and the relation to anatomical structures. He received instructions to program different settings, increase the amplitude by 0.2 mA, and document the resulting effects and side effects. Tremor control was documented by the water-pouring test [10]. Side effects were regarded as permanent if they lasted for more than 30 s. The respective testing results are shown in Figure 2 and Figure 3.

### 2.5. VTA Modeling

The GUIDE XT software, integrated into the Brainlab Elements workflow, was used to calculate the VTA models for both leads and to anticipate (based on imaging) which level of segmented electrodes should ideally be tested to spread current into the different directions. VTA models were calculated for the stimulation settings that resulted in permanent side effects, i.e., sensory face effects (=VPM), sensory hand effects (=VPL), and dysarthria (=internal capsule). The concordance of these models with the observed clinical effects was evaluated based on the respective anatomical segmentation (Figure 2 and Figure 3). A simulation of VTA and the surrounding anatomy was considered a good match with clinical effects if the border of the VTA touched or just crossed the border of the anatomical structure of interest within 0.2 mA (see arrows in Figure 2 and Figure 3).

### 2.6. Ethics

This study was carried out in accordance with the Declaration of Helsinki. The patient gave informed consent to participate in the clinical testing and to publish his clinical and imaging data.

## 3. Results

### 3.1. Lead Location and Orientation

The Euclidean distance between the planned target point and the center of the distal row of segmented electrodes of the implanted lead was 0 mm on the left side and 0.5 mm on the right side towards the patient’s posterior. Based on the immediate postoperative CT, the amount of intracranial air was minimal (0.31 cm^3^), and no brain shift could be observed.

In the midcommissural point coordinates, however, the right target point was located 1.5 mm more anteriorly than the left one. Both leads were positioned similarly in terms of laterality, with a slightly more lateral position of the right lead, which was thus closer to the internal capsule.

Lead orientation was similar for both leads, with the left lead rotated 14° to the right (medial) with respect to the patient’s anterior and the right lead rotated 22° to the left (medial). The leads’ contacts were numbered as follows: left 1–8 and right 9–16 with 2/3/4 being the equivalent of the distal directional contact on the left to 10/11/12 on the right (see Figure 1D).

For each direction the following combinations were activated: anterior left: contact 2 (75%), contact 3 (25%); anterior right: contact 10 (75%), contact 12 (25%). Lateral left: contact 2 (25%), contact 3 (75%); lateral right: contact 10 (25%), contact 12 (75%). Posterior left: contact 3 (50%), contact 4 (50%); posterior right: contact 11 (50%), contact 12 (50%).

### 3.2. Tremor Control

Clinically, the patient benefited well on both sides. For his more severely affected right body side, and hence the left lead, he required 0.8 mA in the water-pouring test [10] to completely control his tremor (i.e., 0/4 with and 2/4 without stimulation). For the left body side, and hence the right lead, the patient experienced complete tremor control at 0.6 mA for this test (i.e., 0/4 with and 1–2/4 without stimulation). Similar effects were seen when steering into either direction.

### 3.3. Side Effects

#### 3.3.1. Sensory Side Effect

On the left body side (right lead), the threshold for permanent sensory side effects was 0.8 mA lower than for the right body side (left lead). When steering 180 degrees away from the VPM/VPL in the anterior direction, sensory side effects occurred at higher currents (2.0 mA vs. 1.8 mA for the left lead, and 1.6 mA vs. 1.0 mA for the right lead). For more details see Figure 2 and Figure 3.

#### 3.3.2. Capsular Side Effects

When steering current laterally towards the internal capsule, the patient developed dysarthria as the first motor side effect. This occurred at 2.8 mA on the slightly more medial left lead, and 2.4 mA on the more lateral right lead. When steering current into the anterior or posterior direction, the necessary threshold was higher. For the left lead, 3.6 mA caused dysarthria. On the right side, the stimulation was stopped at 3.0 mA due to strong sensory side effects, but no dysarthria was observed with this setting. Detailed information is shown in Figure 2 and Figure 3.

### 3.4. VTA Model

Most of the VTAs (12/18) correlated well with the expected side effects. The best concordance was seen for dysarthria, which was anticipated correctly in all three directions on both sides (6/6 VTAs), as well as for sensory side effects of the hand (5/6 VTAs). The least concordance was seen for sensory side effects of the face (1/6 VTAs). The current back spread modeled by GUIDE XT matched segmentation and observed clinical effects for 3/4 VTAs, the current anterior spread for 4/6 VTAs and the current side spread in 5/8 VTAs. In all cases, however, the modeled VTAs seemed slightly too large, in the sense that side effects were anticipated at lower mA settings. This is consistent with the conservative approach implemented by Boston Scientific in GUIDE XT [11].

## 4. Discussion

The main finding of our study is that the imaging software correctly detected the patient’s asymmetrical anatomy, with a more anterior right VIM confirmed by clinical testing using current steering. Despite the more anterior (and thus more distant) location of the implanted right lead from the sensory thalamus, both leads showed similar thresholds for sensory side effects. In fact, according to the automatic segmentation the right lead was actually closer to the sensory thalamus, which correlated with the lower voltage required to cause sensory side effects on that side. This case once again demonstrates how millimeters matter in DBS. A more posterior position of the right lead, as initially planned, would have most likely reduced the clinical benefit, since it would have been positioned more in the sensory thalamus than in the VIM. While such a situation can be detected and adjusted for during awake procedures, it would remain unnoticed during asleep surgery and could potentially necessitate lead revision.

Clinically, the patient benefited well on both sides, even though he was more severely affected on the right body side (left VIM), which makes a direct comparison challenging. Interestingly, despite the smaller amount of current required to reduce tremor on the less-affected left body side, the patient was up-front programmed in the steering mode away from the sensory thalamus; on the right body side, the neurologist did not see the necessity due to a good clinical effect in the ring mode. In theory, a focus “sweet spot” search can be performed with the directional lead; however, in this patient the tremor would have to progress since it was currently well controlled in all modes.

While overall there was a good match between clinical results and the MRI-based segmentation when comparing the right and the left side, the VTA model itself was difficult to evaluate in detail in our case. The VTA model is displayed as a 3D, roughly spherical volume that touches or intersects with various anatomical structures on different levels (*z*-axis). Overall, the VTAs modeled by GUIDE XT in combination with anatomical segmentation showed a good concordance with permanent side effects, especially for dysarthria and sensory side effects of the hand region (VPL). Sensory side effects of the face region (VPM), however, hardly ever matched the simulation, and the VTA size seemed to be overestimated in nearly all cases. When differentiating between different directions of current spread, the back spread, and anterior spread seemed to match segmentation and clinical effects well, while the current side spread exhibited the least concordance. However, no conclusions should be drawn from such observations in this single case, and a lack of concordance could also be explained by an inaccurate or unrealistic border between the VPL and VPM. In this aspect, it is also important to consider that Hassler’s subdivisions of the thalamus are not based on anatomical connectivity, but primarily on histochemical staining, and that the optimal target for tremor might transcend these subdivisions, for example.

Despite its limitations, this case study demonstrates that when using directional DBS leads and visualization software, a thorough understanding of VTA modeling and its accuracy are indispensable. Such an understanding could also ultimately help to adjust planning strategies for directional leads, which might be different from those for standard leads. To date much valuable information has been provided by the open-source research software Lead-DBS [12,13,14,15]. However, like other software products (i.e., PyDBS [16]) it is not CE-marked and was thus not applied or tested in this study. Moreover, one of the biggest advantages of the Brainlab software is the fact that all steps are integrated into the clinical workflow. Thus, they do not require any additional time, software, or skills to perform such evaluations and can provide immediate feedback and valuable information in clinical practice.

Overall, in this patient with asymmetric thalamus anatomy, the imaging software greatly helped to adjust the initial coordinate-based targeting of the VIM and proved to be a valuable addition for surgical planning. It is, however, crucial to also understand its limitations. First of all, any algorithm can only be as good as the images with which it is provided. We therefore used 3T MRI performed under general anesthesia to minimize patient motion, with thin slices and various sequences. Second, any functional neurosurgeon should be able to plan trajectories on raw MRI data without the help of segmentation software. Such atlases should only be used to support traditional planning, but they should never be trusted blindly.

## 5. Conclusions

Automatic anatomical segmentation and VTA modeling software is a promising tool to support targeting as well as postoperative programming in VIM DBS. In particular, in the case of directional DBS leads it is possible to postoperatively assess the concordance of such an image-based simulation with clinical effects in great detail. Understanding the assumptions and limitations underlying VTA modeling, in particular regarding anterior, back, and side spread of current, is an important premise. More data from larger cohorts is required to evaluate the accuracy, reliability, and ultimately the clinical benefit of these software solutions.

## Figures and Tables

**Figure 1 brainsci-10-00642-f001:**
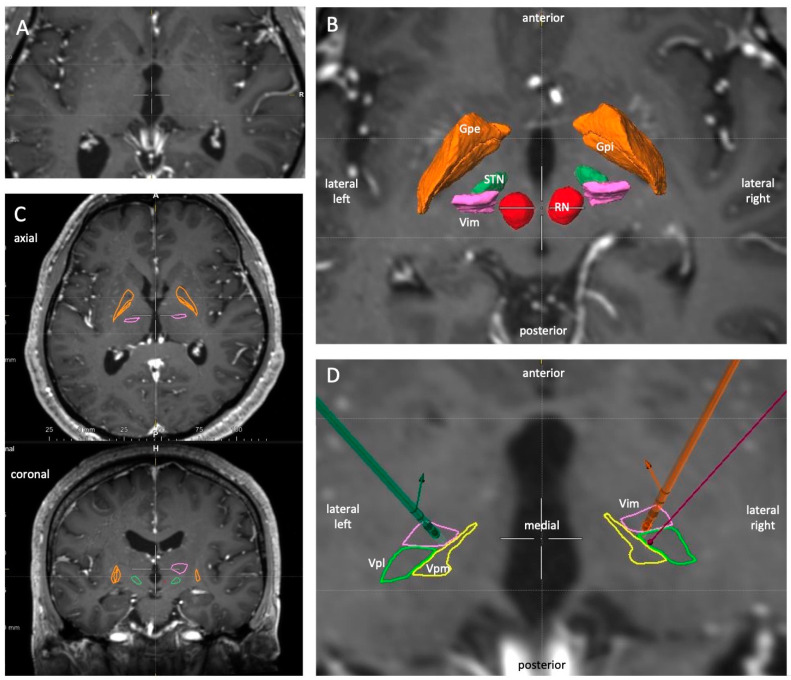
Preoperative planning. (**A**): Axial view of T1 MRI with contrast agent. (**B**): Axial and coronal view of T1 MRI with segmented nuclei. (**C**): Magnification of an axial slice at the level 2 mm above the ACPC level with overlaid segmented structures in 3D: GPi (orange), GPe (orange), STN (green), VIM (pink), and red nucleus (RN, red). Note the asymmetry between the left and right side. (**D**): Further magnification of an axial slice with the segmented VIM, VPL (green), and VPM (yellow) nuclei in 2D. The purple line represents the initially-planned right trajectory, which was then adjusted 2 mm anteriorly. The green and orange leads reflect the implanted positions as localized on postoperative CT and transferred to MRI by rigid fusion. Both leads are rotated slightly medially. 2D, 2-dimensional; 3D, 3-dimensional; ACPC, anterior commissure/anterior commissure; CT, computed tomography; GPi, globus pallidus internus; GPe, Globus pallidus externus; MRI, magnetic resonance imaging; RN, red nucleus; STN, subthalamic nucleus; VIM, ventral intermedius nucleus; VPL, ventral posterior lateral nucleus, VPM, ventral posterior medial nucleus.

**Figure 2 brainsci-10-00642-f002:**
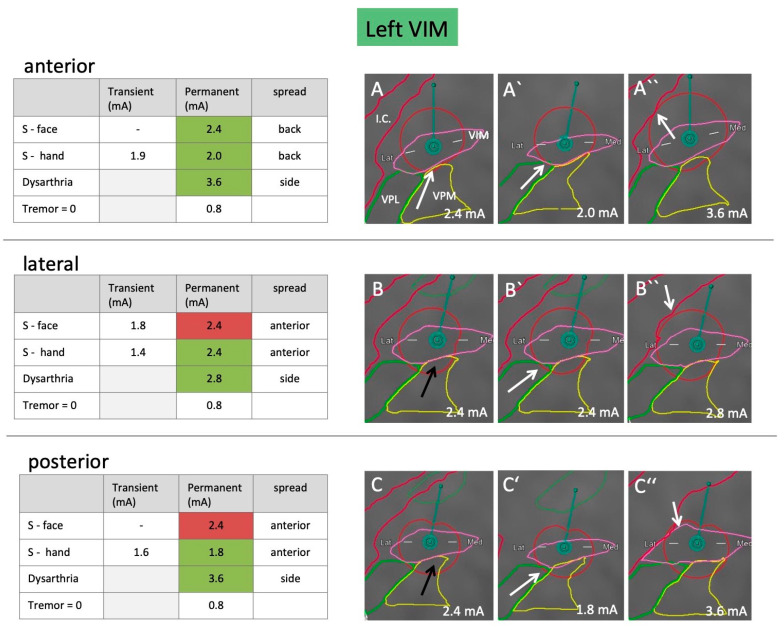
Clinical testing results including side-effect thresholds and volume of tissue activated (VTA) models for the left lead. Testing results show stimulation in the anterior, lateral, and posterior directions using the distal row of segmented electrodes. Denoted is the amount of current (mA) required to provoke transient or permanent sensory side effects in the face (VPM side effect) and hand (VPL side effect), as well as dysarthria (internal capsule side effect). Furthermore, the amount of mA required to reduce tremor to 0 is documented. Table cells highlighted in green reflect VTAs and segmentation, which matched clinical effects well (i.e., first appearance of permanent side effect at the respective voltage), whereas cells highlighted in red reflect those, which did not match well (i.e., permanent side effect was not yet present or had already been observed at lower voltage). Illustration of a predominantly anterior (**A**–**A″**), lateral (**B**–**B″**), and posterior (**C**–**C″**) stimulation to cause facial (**A**–**C**), hand (**A′**–**C′**), and capsular (**A″**–**C″**) side effects as modeled by the software. White arrows pinpoint scrutinized segmentation borders, which in conjunction with the respective VTA seemed plausible, black arrows reflect those which did not. I.C., internal capsule (red); S, sensory; VIM, ventral intermedius nucleus (pink); VPL, ventral posterior lateral nucleus (green); VPM, ventral posterior medial nucleus (yellow).

**Figure 3 brainsci-10-00642-f003:**
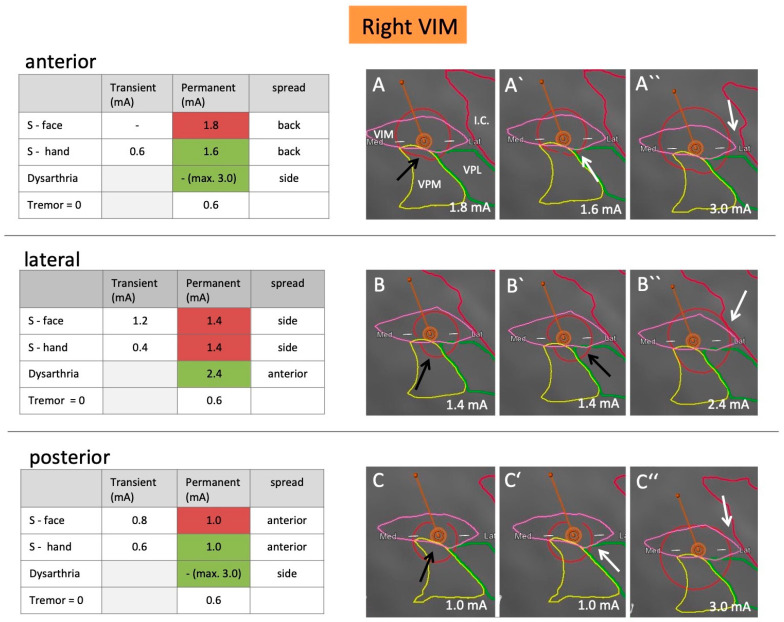
Clinical testing results including side-effect thresholds and VTA models for the right lead. Testing results show stimulation in the anterior, lateral, and posterior directions using the distal row of segmented electrodes. Denoted is the amount of current (mA) required to provoke transient or permanent sensory side effects in the face (VPM side effect) and hand (VPL side effect), as well as dysarthria (internal capsule side effect). Furthermore, the amount of mA required to reduce tremor to 0 is documented. Table cells highlighted in green reflect VTAs and segmentation, which matched clinical effects well (i.e., first appearance of permanent side effect at the respective voltage), whereas cells highlighted in red reflect those, which did not match well (i.e., permanent side effect was not yet present or had already been observed at lower voltage). Illustration of a predominantly anterior (**A**–**A″**), lateral (**B**–**B″**), and posterior (**C**–**C″**) stimulation to cause facial (**A**–**C**), hand (**A′**–**C′**), and capsular (**A″**–**C″**) side effects as modeled by the software. White arrows pinpoint scrutinized segmentation borders, which in conjunction with the respective VTA seemed plausible, black arrows reflect those which did not. I.C., internal capsule (red); S, sensory; VIM, ventral intermedius nucleus (pink); VPL, ventral posterior lateral nucleus (green); VPM, ventral posterior medial nucleus (yellow).

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
