# Peer review of "Evaluation of Automatic Segmentation of Thalamic Nuclei through Clinical Effects Using Directional Deep Brain Stimulation Leads: A Technical Note"

_brainsci, 2020, doi:10.3390/brainsci10090642_

Round 1
Reviewer 1 Report
This is a well-written and relatively straight-forward paper, investigating the utilizing of commercial software to automatically segment sub-thalamic nucleus structures in MRI for deep brain stimulation, and predict which structures will be stimulated based upon volume of tissue activation models and patient side effects.
I commend the authors on the interesting use of side-effect profiles to examine the accuracy of VTA models.
This paper shows that the automatic segmentation provided by Brainlab and the estimation of tissue stimulation by GUIDE XT combined relatively well predict which structures will be activated, based upon induced side effects. However, with the lack of comparison with other algorithms and software implementations, such as those implemented in the open source software Lead-DBS, it is difficult to see exactly how well this methodology performs.
In addition, the automated segmentation of the nuclei, whilst impressive does seem to be validated by manual segmentation (which would not be overly difficult given it is only from one patient), or by the co-registration of other anatomical atlases (e.g. MNI152 or more detailed for the STN, Ewert et al. 2017). In addition VTA, is not compared to other implementations, which may be more accurate.
If the authors would like to make a statement about these pieces of software being accurate it is important that they highlight either their technical superiority to other solutions, or ease of use for hospital setting (including CE certification).
Given that this is a technical paper, and that only one subject is given, I beleive the paper would greatly benefit by comparing methods.
Minor issues: many of the images, which have tissue segmentation, are difficult to see. In particular, given the issues with poor tissue contrast in these structures at least one image without the lines showing the borders next to one with the borders shown would be useful. This would highlight the difficulty of tissue segmentation and the performance of the software.
Author Response
Comment 1: This paper shows that the automatic segmentation provided by Brainlab and the estimation of tissue stimulation by GUIDE XT combined relatively well predict which structures will be activated, based upon induced side effects. However, with the lack of comparison with other algorithms and software implementations, such as those implemented in the open source software Lead-DBS, it is difficult to see exactly how well this methodology performs.
Reply 1: We are aware that there can be significant differences between various segmentation and VTA modeling algorithms, and acknowledge the legitimate question of how other solutions would perform on this particular patient. However, our focus here was on the ease of use and benefit of the CE certified software Brainlab Elements and BSC GUIDE XT in the clinical setting. Moreover we wanted to demonstrate how such software can be evaluated. While e.g. Lead-DBS offers state-of-the-art tools with an impressive list of publications and has been providing interesting new insights in the field of DBS research, it is not a medical device and can hence not be used for surgical planning. Furthermore, the respective workflow is quite cumbersome. We agree that a comparison of various algorithms would be valuable, but we believe such a comparison should always be based on a substantial number of patients and not just a single case in order to allow for a solid conclusion.
Comment 2: In addition, the automated segmentation of the nuclei, whilst impressive does seem to be validated by manual segmentation (which would not be overly difficult given it is only from one patient), or by the co-registration of other anatomical atlases (e.g. MNI152 or more detailed for the STN, Ewert et al. 2017). In addition VTA, is not compared to other implementations, which may be more accurate.
If the authors would like to make a statement about these pieces of software being accurate it is important that they highlight either their technical superiority to other solutions, or ease of use for hospital setting (including CE certification).
Given that this is a technical paper, and that only one subject is given, I believe the paper would greatly benefit by comparing methods.
Reply 2: We agree that we should highlight one of the two aspects. The advantage and superiority of this software is indeed the fact that it is a CE certified product for clinical use. It is an integrated part of the clinical workflow and thus doesn’t require any extra time, computer software or knowledge. This is particularly important and valuable for clinicians, who do not have time to upload and analyze patient’s data in their clinical routine. Since it is used for clinical planning it has to be particularly accurate.
As mentioned above a comparison to other software with a completely different focus and purpose (i.e. research tool vs. planning software) was not the intention of the technical case.
We have added a sentence in the discussion to better emphasize the above mentioned advantages of this CE marked software (page 8, line 242 - 245).
Comment 3: Minor issues: many of the images, which have tissue segmentation, are difficult to see. In particular, given the issues with poor tissue contrast in these structures at least one image without the lines showing the borders next to one with the borders shown would be useful. This would highlight the difficulty of tissue segmentation and the performance of the software.
Reply 3: This is a valid request. We have added an image without the lines at the thalamic level in Figure 1 (now Figure 1 A) on page 3.
Reviewer 2 Report
The authors present a case report where imaging software detected an ET patient’s asymmetrical anatomy with VIM DBS. Additional information and alternative presentation of the results (suggestions below) will add clarity to this study:
- Line 76 - Indicate which directional leads were used here (in methods)
- Line 80 – “he participated in this trial” – consider rewording to “study”; trial implies a clinical trial
- Line 104 – note to fix this error
- Lines 125 – are there any measures of tremor before/after surgery to better define “his tremor was well controlled”?
- Lines 129-130 – include a reference for the water-pouring test
- Lines 147-148 “…matched clinical effects well … did not match well” - please provide a description of criteria used to define matching well vs not well
- Figure 2 – there’s room to include more information in the tables:
- consider adding mA either in table headers or next to each value
- consider spelling out transient and permanent
- maybe use “tremor = 0” instead of “tremor 0” ?
- posterior right VIM should be “-(max. 3.0)”?
- Figure 3 – panels are out of order when viewed top/down; consider renumbering or rearranging; confusing as currently presented
- Line 173 – do you have water-pouring data before surgery? This would put the “benefit” of stimulation into context
- Unsure of journal format requirements, but consider moving 2.6 Ethics section up to the beginning of the methods
- Line 198 – is “the conservative approach implemented by Boston Scientific in GUIDE XT” widely known? May require a reference for this statement; sounds like an opinion (which may be better suited to include in the discussion, not the results)
Author Response
Comment 1: Line 76 - Indicate which directional leads were used here (in methods)
Reply 1: We have added the lead type in the methods on page 2, line 77.
Comment 2: Line 80 – “he participated in this trial” – consider rewording to “study”; trial implies a clinical trial
Reply 2: We agree and have changed the wording accordingly (page 2, line 80)
Comment 3: Line 104 – note to fix this error
Reply 3: We apologize. We refer to Figure 1 D. The purple trajectory shows the initially planned trajectory. The orange electrode was implanted along the adjusted anterior trajectory. We have added a sentence to describe this better (page 3, line 6 of the figure description).
Comment 4: Lines 125 – are there any measures of tremor before/after surgery to better define “his tremor was well controlled”?
Reply 4: Yes, we routinely perform the Fahn-Tolosa Marin Tremor rating scale before and six months after the surgery. We have added the scores of the unilateral tasks and a reference on page 4, lines 126 - 129.
Comment 5: Lines 129-130 – include a reference for the water-pouring test
Reply 5: We have added the reference (page 4, line 132).
Comment 6: Lines 147-148 “…matched clinical effects well … did not match well” - please provide a description of criteria used to define matching well vs not well
Reply 6: Thank you for pointing this out. Matched clinical effect well meant that at the respective voltage the permanent side effect was observed for the first time and did thus match the visualization. Did not match well meant that at the respective voltage the permanent side effect was not yet present or had already been observed at lower voltages and thus did not match the image. We have added this explanation (page 5, lines 149 – 152).
Comment 7: Figure 2 – there’s room to include more information in the tables:
- consider adding mA either in table headers or next to each value
- consider spelling out transient and permanent
- maybe use “tremor = 0” instead of “tremor 0” ?
- posterior right VIM should be “-(max. 3.0)”
Reply 7: We have made the requested changes in the Figure. We have also added 0.6 mA for the anterior right side for Tremor = 0, which was missing.
Comment 8: Figure 3 – panels are out of order when viewed top/down; consider renumbering or rearranging; confusing as currently presented
Reply 8: Thank you for pointing this out. We have rearranged the numbering of his figure and have adapted the legend accordingly. See Figure 3 and ledged on page 6.
Comment 9: Line 173 – do you have water-pouring data before surgery? This would put the “benefit” of stimulation into context.
Reply 9: Yes we do. We have added this data in the manuscript (page 7, line 179 - 181).
Comment 10: Unsure of journal format requirements, but consider moving 2.6 Ethics section up to the beginning of the methods
Reply 10: We believe that this is at the correct place, since the template we used did not suggest otherwise. If not, the editors may move the section to the correct place.
Comment 10: Line 198 – is “the conservative approach implemented by Boston Scientific in GUIDE XT” widely known? May require a reference for this statement; sounds like an opinion (which may be better suited to include in the discussion, not the results)
Reply 10: Thank you for pointing this out. We have added the reference that the conservative BSC VTA modeling is based on in the manuscript on page 7, line 203 (Frankemolle et al., Brain, 2010, now Reference 11).